# Radical Stress Is More Cytotoxic in the Nucleus than in Other Organelles

**DOI:** 10.3390/ijms20174147

**Published:** 2019-08-25

**Authors:** Laurent M. Paardekooper, Ellen van Vroonhoven, Martin ter Beest, Geert van den Bogaart

**Affiliations:** 1Department of Tumor Immunology, Radboud Institute for Molecular Life Sciences, Radboud University Medical Center, 6525 Nijmegen, The Netherlands; 2Department of Molecular Immunology, Groningen Biomolecular Sciences and Biotechnology Institute, University of Groningen, 9747 Groningen, The Netherlands

**Keywords:** reactive oxygen species, oxidative stress, optogenetics, DNA damage

## Abstract

Cells are exposed to reactive oxygen species (ROS) as a by-product of mitochondrial metabolism, especially under hypoxia. ROS are also enzymatically generated at the plasma membrane during inflammation. Radicals cause cellular damage leading to cell death, as they react indiscriminately with surrounding lipids, proteins, and nucleotides. However, ROS are also important for many physiological processes, including signaling, pathogen killing and chemotaxis. The sensitivity of cells to ROS therefore likely depends on the subcellular location of ROS production, but how this affects cell viability is poorly understood. As ROS generation consumes oxygen, and hypoxia-mediated signaling upregulates expression of antioxidant transcription factor Nrf2, it is difficult to discern hypoxic from radical stress. In this study, we developed an optogenetic toolbox for organelle-specific generation of ROS using the photosensitizer protein SuperNova which produces superoxide anion upon excitation with 590 nm light. We fused SuperNova to organelle specific localization signals to induce ROS with high precision. Selective ROS production did not affect cell viability in most organelles except for the nucleus. SuperNova is a promising tool to induce locally targeted ROS production, opening up new possibilities to investigate processes and organelles that are affected by localized ROS production.

## 1. Introduction

Reactive oxygen species (ROS) are a common byproduct of aerobic respiration and lipid metabolism [1,2]. Moreover, ROS can be present at high levels in disease environments due to enzymatic production by NADPH oxidation complexes (NOX) [3] and as a consequence of hypoxia [4,5]. The interplay of ROS with hypoxia is especially relevant. ROS generation leads to local hypoxia by rapid consumption of oxygen [6] and there is strong cross-talk in intracellular signaling in response to ROS and hypoxia. Nrf2 is a major transcription factor upregulated in response to oxidative stress that activates expression of genes carrying a specific antioxidant response element in their promoter [7,8]. However, Nrf2 expression is also controlled by hypoxia-induced factor (HIF), a transcription factor responsive to hypoxia [9]. This makes it difficult to discern between ROS- and hypoxia-induced effects. Since HIF potentiates the Nrf2 signaling pathway and there always is some background generation of ROS, for example from metabolic activity by mitochondria [10], hypoxia may trigger a strong antioxidant reaction. Particularly under inflammatory conditions, ROS synergize with hypoxia to stabilize HIF [11] and upregulate its transcriptional activity [12,13]. In addition, cells are exposed to ROS from the outside environment, e.g., from UV light, radioactive radiation and ozone. ROS and other radicals are generally short-lived species as they rapidly and indiscriminately react with lipids, proteins, and nucleic acids [14,15,16]. Too high ROS levels are well known to trigger cell death via caspase-mediated apoptosis or necrosis [17]. Prolonged elevated ROS levels have widespread effects on general health and, for example, correlate with onset of frailty [18]. Cells evolved molecular mechanisms in order to cope with ROS, including ROS-scavenging antioxidants and ROS neutralizing enzymes [15]. However, ROS are also needed for many physiological processes. In fact, many cellular processes and signaling pathways strongly depend on ROS, like growth factor signaling [19], Src kinase activation [20], chemotaxis [21,22], metabolic feedback regulation [23], induction of autophagy [24], inflammasome formation [25], and tumor necrosis factor (TNF)-α induced apoptosis [26]. In the immune system, ROS have many important functions, including killing of pathogens, antigen degradation and inflammatory signaling to other cells of the immune system [7,27,28,29,30]. ROS are also a key regulator of antigen (cross-)presentation [31,32,33]. In order to allow these physiological functions, cells should not completely degrade and/or sequester ROS and have to cope with a certain level of radical stress.

Both the generation and functioning of ROS occur at distinct subcellular sites. Due to their roles in aerobic respiration and lipid oxidation, mitochondria are exposed to relatively high levels of ROS [34,35]. In immune cells and endothelial cells, NOX complexes can generate large amounts of ROS at the plasma membrane and in the lumen of endosomes and phagosomes [3]. Comparatively little is known about ROS in the endoplasmic reticulum (ER) or Golgi compartments, but it is suggested that NOX activity and the concomitant change in pH play a role in disulfide bond formation during protein folding in the ER, as well as glycan alterations and Ca^2+^ homeostasis in the Golgi apparatus [36,37]. The nucleus can be expected to be exposed to comparatively low amounts of ROS, as external ROS have to diffuse first through the cytoplasm where they can react or be scavenged by antioxidant mechanisms. Additionally, the nucleus is protected by superoxide dismutase 1 [15]. Thus, different organelles are exposed to different levels of ROS and the sensitivity of cells to ROS-induced cell death likely depends on the organelle that is exposed to ROS. However, in most experiments addressing ROS, ROS are induced systemically (i.e., through the entire cell) by addition of compounds such as *tert*-butyl hydroperoxide (TBHP), arsenic or iron to the culture medium. However, since the effects of ROS likely depend on the subcellular location of ROS generation, a method that allows for targeted production of ROS would be a valuable addition to the field.

In this study, we developed an optogenetic toolkit to induce ROS with organellar precision. We used SuperNova, a monomeric genetically-encoded photosensitizing fluorescent protein derived from KillerRed, which shows improved localization and does not perturb mitotic cell division [38,39,40]. The open structure of the β-barrel of SuperNova creates a water channel which connects to the chromophore [39]. ROS (singlet oxygen and superoxide anion) are generated upon excitation of this chromophore and can subsequently exit through the water channel. We generated constructs coding for SuperNova fusion proteins targeted to mitochondria, endosomes, *trans-*Golgi network, nucleus and ER. We verified the localization of these SuperNova fusion proteins to their target organelles by microscopy. The effects of localized ROS production on cell viability were assessed to compare the effects or radical stress in specific organelles. We found that cell viability was only affected upon ROS induction in the nucleus, but not in other organelles, indicating that cells are particularly sensitive to ROS in the nucleus. Our results show that our SuperNova fusion protein toolbox is a viable method of inducing ROS with organellar precision and may aid in investigating the influence of hypoxia on localized antioxidant responses.

## 2. Results

### 2.1. Targeting SuperNova to Specific Organelles

In order to enable the induction of radical stress at specific organelles, we targeted SuperNova to designated subcellular sites by fusing it to proteins and targeting sequences that locate to specific organelles (Figure 1). We fused SuperNova to the mitochondrial protein cytochrome c oxidase subunit 8A (COX8), the luminal site of the endosomal protein vesicle associated membrane protein 8 (VAMP8), the trans-Golgi network integral membrane protein 3 (TGON3), the signal sequence of the ER protein calreticulin and to a nuclear localization signal (NLS) of human c-myc (PAAKRVKLD) [41].

To validate the correct localization of these SuperNova fusion proteins to the targeted organelles, we performed confocal microscopy experiments in HeLa cells. In addition to the plasmids coding for the SuperNova fusion proteins, mCherry-N1 and SuperNova-N1 plasmids (i.e., empty backbone vectors) were used as controls, which code for untagged mCherry and SuperNova, respectively, that locate uniformly in the nucleus and cytoplasm. The excitation and emission spectra of mCherry overlap with those of SuperNova, but mCherry less efficiently induces ROS production upon excitation than SuperNova [38]. Counterstains with organelle-specific antibodies were applied to confirm the localization of the SuperNova fusion proteins. For all SuperNova constructs, they correctly localized to the target organelles (Figure 2A), which was quantified by determination of the Pearson’s colocalization coefficient between the SuperNova fluorescence and the antibody staining (Figure 2B). For VAMP8-SuperNova, the colocalization with early endosomal marker EEA1 was relatively low, because VAMP8 is also present on late endosomes [42]. Likewise, the colocalization of SuperNova-TGON3 with giantin was relatively low due to giantin being a marker of cis- and medial-Golgi apparatus [43,44], whereas TGON3 localizes to medial- and trans-Golgi apparatus [45]. KDEL-tagged constructs are retrieved from cis-Golgi apparatus to the ER and therefore not only localize to the ER but also partially to the Golgi apparatus [46], which can explain the comparatively lower Pearson’s R value of ER-SuperNova.

### 2.2. Nuclear ROS Production Induces Cell Death

SuperNova and its precursor KillerRed are known to be able to induce cell death in a variety of mammalian cell lines and *Caenorhabditis elegans* [40,47,48]. However, in these studies, they used untagged probes that uniformly distributed through the cell or were targeted to mitochondria. In this study, we compared the effects on cell viability upon radical stress induced by SuperNova targeted to other organelles. First, we determined the transfection efficiencies using flow cytometry for constructs coding for SuperNova targeted to mitochondria, endosomes, the trans-Golgi network, the nucleus and the ER by means of fusion proteins. A gate for viable COS-7 cells was set using untransfected COS-7 cells (Figure 3A), after which the percentage SuperNova-positive cells within this gate was determined for each sample (Figure 3B). Depending on the experiment and the construct, we achieved a transfection efficiency of 55%–80% in COS-7 cells (Figure 3C). We then evaluated the effects of targeted ROS production on cell viability by using flow cytometry with Zombie Violet, a fixable live-dead staining. Zombie Violet is an amine-reactive fluorescent dye that is only permeable in cells with compromised membranes (Figure 3D). The transfected COS-7 cells were exposed to 590 nm light for 24 h to activate SuperNova. The cells were stained directly after light exposure. Compared to untransfected cells, exposure to 590 nm light was able to induce cell death for all constructs and this was significant for SuperNova targeted to endosomes and the nucleus (Figure 3E). Cell viability was already reduced at 1 h light exposure (the shortest time-point addressed) and did not significantly differ from 24 h illumination, in line with the finding that SuperNova and KillerRed can rapidly induce cell death within minutes to hours after light exposure [38,40]. Illumination of SuperNova targeted to the nucleus also significantly increased cell death compared to mCherry-N1, which was used to evaluate the effect of ROS production by conventional fluorescent proteins. When compared to the dark condition, light exposure significantly induced cell death for all SuperNova constructs, except for SuperNova targeted to the ER (Figure 3E,F). SuperNova-induced ROS production caused the largest increase in cell death when it was targeted to the nucleus (Figure 3E), despite having a lower transfection efficiency than mCherry or cytosolic SuperNova (Figure 3C).

Surprisingly, NLS-SuperNova already significantly reduced cell viability in absence of 590 nm light exposure (dark condition; Figure 3E). A possible explanation for this is that the exposure of NLS-SuperNova by the ambient (background) light in the cell culture room already sufficed for induction of cell death. Although we took ample precautions to avoid unintended light exposure of the cells in our experiments (culturing in dim conditions and covering of the cells whenever possible), tube lights have dominant emission at the excitation peak of SuperNova (540–550 nm) and therefore might have triggered cell death. In this case, at least a population of the cells would be highly sensitive to light-induced cell death. As an alternative explanation for the loss of cell viability in the dark condition, perhaps SuperNova can interact with a nuclear factor already in absence of light and this might induce cell death. However, arguing against this explanation is the fact that expression of untargeted SuperNova, which is small enough to passively enter the nucleus, did not affect cell viability (Figure 3E). In any case, the excitation of NLS-SuperNova resulted in a significant loss of cell viability compared to the dark condition (Figure 3E,F), supporting our conclusion that nuclear light-induced radical stress induces cell death.

## 3. Discussion

In this study, we compared the effects on cell viability of radical stress at different organelles. We developed an optogenetic toolkit to induce ROS with organellar precision based on the superoxide-producing photosensitizing protein SuperNova fused to various organellar localization motifs [38]. We found that excitation of SuperNova caused significant cell death in cells when it was targeted to the nucleus, whereas targeting SuperNova to other organelles did not strongly affect cell viability. This study contrasts other studies where activation of mitochondrial and untargeted SuperNova and KillerRed were found to induce cell death [38,40,47] and this discrepancy may be caused by a difference in excitation light intensity and/or cell type. The nucleus might be more sensitive to radical stress than other organelles, as ROS can cause DNA damage and many antioxidant mechanisms are preferentially located in the cytosol and mitochondria [15]. Additionally, nuclear ROS may directly induce Bax-mediated apoptosis via activation of nucleophosmin [49,50].

As mentioned in the introduction section, there is a large overlap in antioxidant (Nrf2) and hypoxic (HIF) signaling. Our optogenetic toolkit allows to better distinguish between these effects by producing ROS within a target organelle without affecting the entire cell. An additional advantage of SuperNova is that it yields great temporal control of ROS production at specific organelles [47,48,51]. To date, the roles of ROS in endosomes [27,52] and mitochondria [7,27,53] are quite well studied, but knowledge of their impact on other organelles is still limited. In addition, radical-inducing proteins such as SuperNova provide new possibilities for the development of animal models for studying ROS in diseases. Because SuperNova allows to kill specific cell types (using specific promotors), induce radical stress at specific locations (using localized excitation light) and trigger sterile inflammation, it offers new opportunities to study disease mechanisms, for example anti-cancer immunity and early onset mechanisms of autoimmune diseases. Shirmanova et al. used KillerRed fused to the nuclear protein histone 2B as well as to a mitochondrial targeting motif to study the effects of localized ROS production in cancer radiation therapy in a mouse xenograft tumor model [54]. Teh et al. developed zebrafish models expressing membrane-tagged KillerRed in the hindbrain and in the heart [55]. Shibuya et al. demonstrated slower development of *C. elegans* larvae that expressed mitochondria-targeted KillerRed in muscle tissue [48]. Williams et al. also developed a *C. elegans* model, but now expressed KillerRed specifically in neurons [49]. In this study, activation of untargeted KillerRed resulted in neuronal degeneration and cell death, whereas ROS production in mitochondria only caused organelle fragmentation but did not affect viability [49]. Our data now also indicates that the targeting of optogenetic sensitizers to the nucleus might be the most effective approach for light induction of cell death.

## 4. Materials and Methods

### 4.1. Cloning

The SuperNova plasmid was a gift from Takeharu Nagai (Addgene plasmid #53234). The construct for cytosolic expression of SuperNova was constructed by replacing EGFP in pEGFP-N1 with synthetic SuperNova fused to a *C*-terminal Myc-tag using restriction sites BamHI and NotI. Mitochondrial targeting was achieved by inserting COX8A into the cytosolic SuperNova vector using restriction sites XhoI and HindIII thereby tagging the *C*-terminus of COX8A with SuperNova. Endosomal targeting was achieved by inserting VAMP8 into the cytosolic SuperNova vector using restriction sites HindIII and BamHI, tagging the *C*-terminus of VAMP8 with SuperNova which results in luminal localization of SuperNova. *Trans-*Golgi network targeting was achieved by inserting synthetic DNA coding for TGON3 with *N*-terminal SuperNova into pcDNA3.1 (+) using restriction sites NheI and XbaI. Targeting of the ER was achieved by inserting synthetic DNA coding for the signal sequence of calreticulin (first 18 amino acids) with *C*-terminal SuperNova into pcDNA3.1 (+) using restriction sites HindIII and XbaI. Additionally, an ER retention signal (KDEL) was added to the *C*-terminus of SuperNova. Nuclear targeting was achieved by replacing EGFP in pEGFP-C1 with synthetic DNA coding for SuperNova with an *N*-terminal c-myc nuclear localization signal (NLS) using restriction sites NheI and HindIII. Additionally, the vectors encoding COX8A-, VAMP8-, NLS- and empty vector SuperNova feature a myc-tag at the SuperNova *C*-terminus. All sequences and plasmid maps are shown in the Appendix A. Plasmids have been deposited at Addgene.

### 4.2. Cells, Transfection, and ROS Induction

All experiments were performed in COS-7 cells (ATCC, Manassas, VA, USA; ref# CRL-1651), except for the localization experiments which were performed in HeLa (ATCC; ref# CCL-2). COS-7 cells were cultured in complete DMEM containing glutamine, 10% fetal bovine serum and 1% Antibiotic-Antimycotic. HeLa cells were cultured in complete RPMI containing 10% fetal bovine serum and 1% Antibiotic-Antimycotic. Constructs mCherry-N1, SuperNova-N1, COX8-SuperNova, VAMP8-SuperNova, SuperNova-TGON3, NLS-SuperNova and calreticulin-SuperNova-KDEL were transfected of HeLa or COS-7 cells using Lipofectamine 3000 (Invitrogen by Thermo Fisher Scientific, Waltham, MA, USA; ref# 3000-015) in Opti-MEM Reduced Serum Medium (Life technologies by Thermo Fisher Scientific, Carlsbad, CA, USA; ref# 11058-021). To activate SuperNova, we cultured cells in phenol-red free culture media and placed them under a 1.4 mW/cm^2^ 590 nm LED array (LIU590A, Thorlabs, Newton, NJ, USA) with diffuser (DG20-600, Thorlabs) at 37°C, 5% CO_2_. 

### 4.3. Microscopy

Localization experiments were performed in HeLa cells (5 × 10^4^ cells/glass) 24 h post-transfection seeded on ethanol-sterilized 12 mm microscopy glasses. 500 nM MitoTracker Deep Red FM (Thermo Fisher Scientific, Waltham, MA, USA; cat# M22426) was added to COX8-SuperNova transfected cells prior to fixation. Samples were stained as described previously [56] with mouse monoclonal anti-Myc (sc-40; Santa Cruz, Santa Cruz, CA, USA), rabbit polyclonal anti-EEA1 (610456, BD Biosciences, Franklin Lakes, NJ, USA), mouse monoclonal anti-Giantin (ALX-804-600, Enzo, Farmingdale, NY, USA), mouse monoclonal anti-PDI (NB300-517; Novus Bio, Centennial, CO, USA) (all at 1:100 dilution *v*/*v*, except for anti-Giantin 1:500 *v*/*v*) in combination with donkey anti-mouse Alexa488 (A21202; Thermo Fisher), donkey anti-rabbit Alexa647 (A31573; Thermo Fisher), donkey anti-mouse Alexa647 (A31571; Thermo Fisher) or donkey anti-rabbit Alexa647 (A21447; Thermo Fisher) (all at 1:400 dilution *v*/*v*). The cells were fixed with mounting medium containing 4′,6-diamidino-2-phenylindole (DAPI; 0.1 μg/mL), 0.01% Trolox (6-hydroxy-2,5,7,8-tetramethylchroman-2-carboxylic acid) and 68% glycerol in 200 mM sodium phosphate buffer at pH 7.5 and imaged on a Leica SP8 confocal microscope (Leica Microsystems, Wetzlar, Germany) using a Leica HC PL APO CS2 63x/1.2 water immersion objective.

### 4.4. Flow Cytometry

Transfection efficiencies and cell viabilities were determined in COS-7 cells. Cells were transfected with mCherry and SuperNova constructs in a 6-well plate (2.5 × 10^5^ cells/well), trypsinized 24 h post-transfection, washed and resuspended in 100 µL PBS for flow cytometry analysis with a BD FACSVerse flow cytometer (BD Biosciences). Untransfected cells were used to apply gating and to discriminate between SuperNova-positive and negative cells. For cell viability evaluation, the culture medium of the cells was changed to phenol-red free prior to illumination of the SuperNova. SuperNova was activated with 590 nm light for 24 h. Cells were collected in 96-wells v-bottom plates by trypsinization after treatment and stained with Zombie Violet fixable viability dye (BioLegend, San Diego, CA, USA; cat# 423113; 1:2000 dilution *v*/*v*). Cells were fixed in 4% PFA for 5 min, washed with PBS containing 1% BSA and 0.05% sodium azide (PBA) and resuspended in 100 µL PBA for flow cytometry analysis with a BD FACSVerse flow cytometer (BD Biosciences). Flow cytometry data was analyzed using FlowJo X (FlowJo LLC, Ashland, OR, USA).

### 4.5. Statistical Analysis

The data of the transfection efficiency assay was analyzed using a paired one-way ANOVA with post-hoc Tukey’s multiple comparisons test. The data of the viability assay was analyzed using a one-way ANOVA with a post-hoc Dunnett’s multiple comparisons test. Two-sided *p* values < 0.05 were considered to be statistically significant (* *p* < 0.05, ** *p* < 0.01, *** *p* < 0.001).

## 5. Conclusions

The monomeric genetically-encoded photosensitizing fluorescent protein SuperNova was targeted to mitochondria, endosomes, the trans-Golgi network, the nucleus and the ER by means of fusion proteins. Illumination of these SuperNova fusion proteins expressed in COS-7 cells resulted in most cell death when SuperNova was targeted to the nucleus compared to the other organelles. This finding indicates that cells are particularly sensitive to ROS in the nucleus. The SuperNova fusion protein toolbox enables to induce ROS with organellar precision and may aid in investigating the influence of hypoxia on localized antioxidant responses.

## Figures and Tables

**Figure 1 ijms-20-04147-f001:**
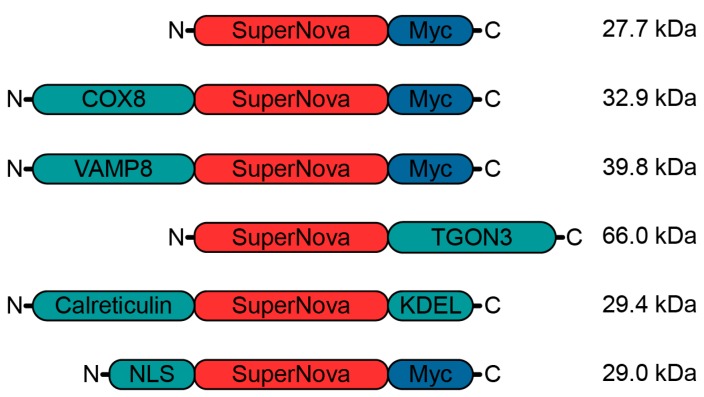
Schematic overview of SuperNova fusion proteins. Myc: myc-tag; COX8: mitochondrial protein cytochrome c oxidase subunit 8A; VAMP8: endosomal vesicle associated membrane protein 8; TGON3: trans-Golgi network integral membrane protein 2; Calreticulin: endoplasmic reticulum (ER) protein; KDEL: ER-retention signal; NLS: nuclear localization signal. Molecular weights of the fusion constructs are indicated.

**Figure 2 ijms-20-04147-f002:**
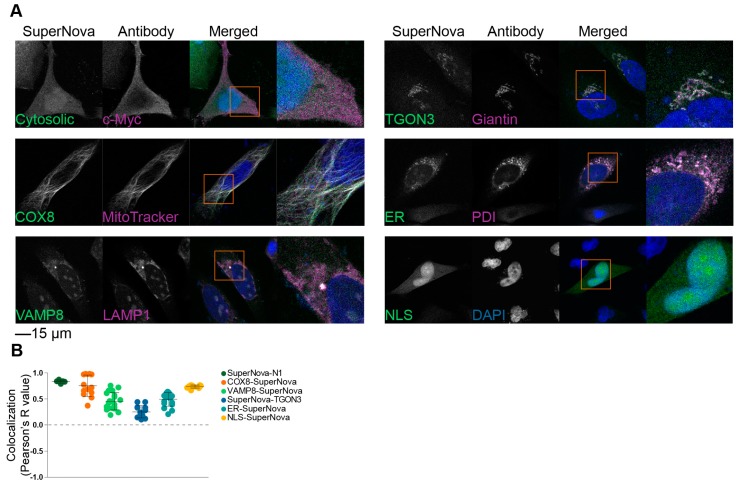
Localization of SuperNova fusion constructs. (**A**) Representative confocal images of HeLa cells transfected with constructs encoding for SuperNova fusion proteins with appropriate counter stains for the target organelles. DAPI is in blue. Cytosolic: SuperNova-N1. Scale bar: 15 µm. (**B**) Pearson’s colocalization coefficients of the SuperNova fusion proteins with the organellar markers from panel **A**. Three technical repeats, each dot represents a microscopy image.

**Figure 3 ijms-20-04147-f003:**
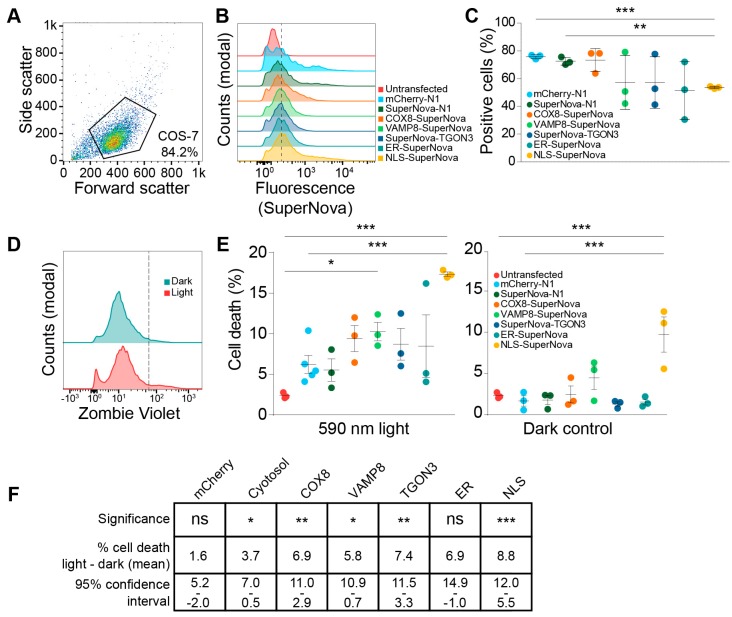
SuperNova fusion construct expression levels and viability upon light exposure. (**A**) Representative scatter plot of COS-7 cells. Polygon indicates gating to exclude debris from cells. (**B**) Fluorescent intensity histograms of SuperNova in COS-7 cells expression SuperNova fusion proteins after gating as shown in **A**. Dotted line indicates gating for SuperNova-positive cell population based on background fluorescence of untransfected cells. At least 20,000 cells were included in each measurement. (**C**) Percentage of COS-7 cells positive for SuperNova as shown in **B**, each dot represents a technical repeat. Analyzed with paired one-way ANOVA with Tukey post-hoc test (** *p* < 0.01; *** *p* < 0.001). (**D**) Representative fluorescent intensity histograms of COS-7 cells expressing SuperNova fusion protein (cytosolic) after stimulation with 590 nm excitation light for 1, 4, or 24 h and stained with the Zombie Violet cell viability dye. Dotted line indicates gating for Zombie Violet-positive cell population based on background fluorescence of unstained cells (not shown). (**E**) Percentage of cell death in COS-7 after stimulation with 590 nm excitation light for 24 h, each dot represents a technical repeat. Control cells were kept in the dark for 24 h. At least 20,000 cells were included in each measurement. Analyzed with unpaired one-way ANOVA with Dunnett post-hoc test comparing all conditions to the untransfected cells and to mCherry-N1 (* *p* < 0.05; *** *p* < 0.001). (**F**) Table showing analysis of data from panel **E**. Confidence intervals shown are 95% of average difference in cell death (%) between dark and 590 nm light exposure. Analyzed with unpaired one-way ANOVA with Dunnett post-hoc test (ns: not significant; * *p* < 0.05; ** *p* < 0.01; *** *p* < 0.001).

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
