# Peer review of "Radical Stress Is More Cytotoxic in the Nucleus than in Other Organelles"

_ijms, 2019, doi:10.3390/ijms20174147_

Round 1

Reviewer 1 Report

This paper developed an optogenetic toolbox for organelle-specific generation of ROS using the photosensitizer protein SuperNova. They investigeted that cells are more sensitive to ROS production in the nucleus compared to other organelles, likely due to DNA damage.

This paper might be interested for some readers in optogenetics.

Some suggestions:

1) the title sounds over absolute, may think about another reserved expression.

2) the overall structure may need to be improved. After the introduction section, it directly jumped into Results session, which is quite confusing. may think about move the Methods and materials section up.

3) A conclusion section may be required then at the end reader can clearly understand the main findings of the work

4) considering as a full journal article,  the content may not be sufficient.

5) there are several grammatical errors and typos need to corrected.

Author Response

This paper developed an optogenetic toolbox for organelle-specific generation of ROS using the photosensitizer protein SuperNova. They investigeted that cells are more sensitive to ROS production in the nucleus compared to other organelles, likely due to DNA damage.

This paper might be interested for some readers in optogenetics.

Some suggestions:

1.1) the title sounds over absolute, may think about another reserved expression.

We changed the title to: “Radical stress is more cytotoxic in the nucleus than in other organelles”.

1.2) the overall structure may need to be improved. After the introduction section, it directly jumped into Results session, which is quite confusing. may think about move the Methods and materials section up.

We moved the Material and Methods section to before the Results, as suggested by the reviewer.

1.3) A conclusion section may be required then at the end reader can clearly understand the main findings of the work

As requested by the reviewer, we have added a conclusion section at the end of our manuscript (section 5).

1.4) considering as a full journal article,  the content may not be sufficient.

We believe that more experiments might distract from the main and simple message of the paper: targeting ROS in the nucleus is more cytotoxic than other organelles. Please note that we chose to focus on cell viability, given the interest in SuperNova (and other optogenetics photosensitizers) in animal models, where they are used to remove distinct cell types (see Discussion section). We will deposit all constructs at the non-profit repository Addgene (https://www.addgene.org/Geert_van_den_Bogaart/).

1.5) there are several grammatical errors and typos need to corrected.

We carefully proof-read our manuscript and corrected the language.

Reviewer 2 Report

In their manuscript “Radical stress is more detrimental in the nucleus than in other organelles” Laurent Paardekooper and co-authors describe an elegant method to induce oxidative stress in defined cellular sub-localizations, using organelle-targeting of a fluorescent protein named SuperNova.

The central technique using superoxide anion production by SuperNova appears extremely promising, and the study design is well suited to address the impact of ROS in various cellular compartments. The manuscript itself is very well written and presents both experimental procedures and results clearly

While I think that the main purpose of this manuscript is the introduction of the technique (which is done excellently), I’d like to raise some critics regarding the interpretation of the results:

The authors find that SuperNova in the nucleus shows most deleterious effects and induces cell death, while other subcellular localizations affect cell survival only marginally. In this context Paardekooper et al. fail to discuss that nuclear SuperNova induces cell death also without light exposure and therefore without ROS production (Fig 3E right panel), and it is unclear whether there is any statistical difference between the light exposed and dark control NLS-SuperNova cells at all.

Therefore I hope the authors can include more control experiments addressing this nuclear effect. Interesting would be e.g. a time course of Zombie Violet staining/cell death like in 3D for nuclear SuperNova. Important would be the direct comparison of light-exposed and dark-control cells – is there any statistical difference?

If there is no way to perform additional experiments, it might be sufficient to evaluate the existing data of Fig. 3E. Currently the highly significant difference of NLS-SuperNova to mCherry is described, but it is unclear whether everything else is nonsignificant. Especially the difference between NLS-SuperNova and mCherry in the dark control looks quite impressive, too.

Author Response

In their manuscript “Radical stress is more detrimental in the nucleus than in other organelles” Laurent Paardekooper and co-authors describe an elegant method to induce oxidative stress in defined cellular sub-localizations, using organelle-targeting of a fluorescent protein named SuperNova.

The central technique using superoxide anion production by SuperNova appears extremely promising, and the study design is well suited to address the impact of ROS in various cellular compartments. The manuscript itself is very well written and presents both experimental procedures and results clearly

While I think that the main purpose of this manuscript is the introduction of the technique (which is done excellently), I’d like to raise some critics regarding the interpretation of the results:

2.1) The authors find that SuperNova in the nucleus shows most deleterious effects and induces cell death, while other subcellular localizations affect cell survival only marginally. In this context Paardekooper et al. fail to discuss that nuclear SuperNova induces cell death also without light exposure and therefore without ROS production (Fig 3E right panel), and it is unclear whether there is any statistical difference between the light exposed and dark control NLS-SuperNova cells at all.

We agree that NLS-SuperNova already reduced cell viability (albeit less) in the absence of the 590 nm light exposure (dark control). An explanation for this is that the exposure of NLS-NuperNova to the ambient (background) light in our cell culture room already sufficed for induction of cell death. We now discuss this possibility in our revised manuscript (page 6, line 216-219). We also indicate significance for all relevant conditions (see point #2.3 below).

2.2) Therefore I hope the authors can include more control experiments addressing this nuclear effect. Interesting would be e.g. a time course of Zombie Violet staining/cell death like in 3D for nuclear SuperNova. Important would be the direct comparison of light-exposed and dark-control cells – is there any statistical difference?

We attempted a time course, but were unable to capture the kinetics of cell death, likely because this is a very fast event and probably not limited by the rate of ROS production but rather by the time required for the cells to become permeable for the Zombie Violet viability dye. Such fast kinetics are in line with our finding that the dark-control cells already showed considerable cell death, as explained above at point #2.1. We now comment on this in our revised manuscript (page 6, line 216-219).

The light-exposed and dark-control cells were already performed side-by-side, and was found to be significant. We now provide full statistical analysis as new figure 3F and discuss this in our revised manuscript (page 6, line 215-216).

2.3) If there is no way to perform additional experiments, it might be sufficient to evaluate the existing data of Fig. 3E. Currently the highly significant difference of NLS-SuperNova to mCherry is described, but it is unclear whether everything else is nonsignificant. Especially the difference between NLS-SuperNova and mCherry in the dark control looks quite impressive, too.

We now show the significance of all relevant conditions in figure 3F and expanded our evaluation of the results. See point #2.1 above.

Round 2

Reviewer 1 Report

The authors have addressed all my suggestions. Suggest for publication. 

Reviewer 2 Report

In their resubmitted manuscript Paardekooper et al. have addressed the one big critical point that I had with the initial manuscript version, as they now analyze and discuss the obvious cytotoxicity of nculearly localized Supernova without intentional light exposure.

I am not very convinced by the answers, though: if the ambient light during cell culture is strong enough to induce significant cell death, it would have been wise to adapt the experimental conditions - cell culture is usually done in very dim light when dealing with photosensitive substances, so that would have been appropriate here, too.

Also the observation that cell death is happening too fast to be observed is a bit doubtful: even the time course cell death by extreme oxidative stress (e.g. 100 uM H2O2) can be followed, so if it is not possible here, it speaks for the fact that most cells are dead before even starting the light exposure.

Author Response

REVIEWER 2
In their resubmitted manuscript Paardekooper et al. have addressed the one big critical point that I had with the initial manuscript version, as they now analyze and discuss the obvious cytotoxicity of nculearly localized Supernova without intentional light exposure.
I am not very convinced by the answers, though: if the ambient light during cell culture is strong enough to induce significant cell death, it would have been wise to adapt the experimental conditions - cell culture is usually done in very dim light when dealing with photosensitive substances, so that would have been appropriate here, too.

We agree with the Reviewer that it seems not very likely that the low ambient light sufficed for the killing of the cells, but this still is the most plausible explanation we can think of. In our experiments, we took ample precautions to avoid unintended light exposure and cultured the cells in dim conditions (with the light in the laminar flow cabinet switched off) and covered the cells with tin foil whenever possible. However, we still observed cell death in the ‘dark’ condition for NLS-SuperNova, which we did not observe for the other constructs. Please note that tube lights have dominant emission at 540-550 nm, which is the excitation peak of SuperNova, and the ambient light thus still might have triggered cell death. In this case, at least a population of the cells would be highly sensitive to light-induced cell death. We have added this explanation to our manuscript (page 5). In addition, we now provide the alternative explanation for the loss of cell viability in the dark condition that SuperNova perhaps interacts with a nuclear factor already in absence of light and this might induce cell death. However, arguing against this explanation is the fact that the other SuperNova constructs, including untargeted NuperNova which is small enough to passively enter the nucleus, did not affect cell viability. In any case, the excitation of NLS-SuperNova resulted in a significant loss of cell viability compared to the dark condition, supporting our conclusion that nuclear light-induced radical stress induces cell death. In our revised manuscript, we rewrote our explanation of this observation and now provide the arguments above (page 5; marked in yellow).

Also the observation that cell death is happening too fast to be observed is a bit doubtful: even the time course cell death by extreme oxidative stress (e.g. 100 uM H2O2) can be followed, so if it is not possible here, it speaks for the fact that most cells are dead before even starting the light exposure.

We agree with the Reviewer that SuperNova-induced cell death can be followed over time and it has been shown previously that SuperNova and KillerRed-induced cell death occurs at the minutes-hour time scale [Takemoto, et al. (2013) Sci. Rep. 3:2629; Bulina, et al. (2006) Nat. Biotechnol. 24:95]. However, please note that the light-induced radical formation is only the trigger of cell death, but that subsequent events such as caspase activation and permeability of the cell membrane are probably rate-limiting for this event. We rewrote this section (page 4-5) and now explain that we observed reduced cell viability already at 1 hr light exposure and this did not significantly differ from 24 hr illumination. We did not address shorter time scales, as the purpose of this study was a side-by-side comparison of the cytotoxicity of SuperNova targeted to different organelles under identical exposure conditions (same duration of exposure, light intensity and uniformity of field of illumination). Because cell death increased significantly compared to the dark condition, we can exclude that the cells were dead before starting the light exposure.

Round 3

Reviewer 2 Report

As the authors have now clearly described their experimental conditions and also openly discuss the weak point of cell death in non-exposed nuclear SuperNova cells, my critics are answered and I consider the manuscript now to be fine.